# Operationalizing the 2018 World Cancer Research Fund/American Institute for Cancer Research (WCRF/AICR) Cancer Prevention Recommendations: A Standardized Scoring System

**DOI:** 10.3390/nu11071572

**Published:** 2019-07-12

**Authors:** Marissa M. Shams-White, Nigel T. Brockton, Panagiota Mitrou, Dora Romaguera, Susannah Brown, Alice Bender, Lisa L. Kahle, Jill Reedy

**Affiliations:** 1Risk Factor Assessment Branch, Epidemiology and Genomics Research Program, Division of Cancer Control and Population Sciences, National Cancer Institute, Bethesda, MD 20892, USA; 2American Institute for Cancer Research, Arlington, VA 22209, USA; 3World Cancer Research Fund International, 22 Bedford Square, London, WC1B 3HH, UK; 4Balearic Islands Health Research Institute (IdISBa), University Hospital Son Espases, 07120 Palma de Mallorca, Spain; 5Barcelona Institute for Global Health (ISGlobal), 08003 Barcelona, Spain; 6CIBER Physiopathology of Obesity and Nutrition (CIBEROBN), 28029 Madrid, Spain; 7Information Management Services, Inc., Rockville, MD 20850, USA

**Keywords:** cancer prevention, diet, weight, physical activity, breastfeeding, dietary guidelines, index score

## Abstract

Background: Following the publication of the 2018 World Cancer Research Fund (WCRF) and American Institute for Cancer Research (AICR) Third Expert Report, a collaborative group was formed to develop a standardized scoring system and provide guidance for research applications. Methods: The 2018 WCRF/AICR Cancer Prevention Recommendations, goals, and statements of advice were examined to define components of the new Score. Cut-points for scoring were based on quantitative guidance in the 2018 Recommendations and other guidelines, past research that operationalized 2007 WCRF/AICR Recommendations, and advice from the Continuous Update Project Expert Panel. Results: Eight of the ten 2018 WCRF/AICR Recommendations concerning weight, physical activity, diet, and breastfeeding (optional), were selected for inclusion. Each component is worth one point: 1, 0.5, and 0 points for fully, partially, and not meeting each recommendation, respectively (Score: 0 to 7–8 points). Two recommendations on dietary supplement use and for cancer survivors are not included due to operational redundancy. Additional guidance stresses the importance of accounting for other risk factors (e.g., smoking) in relevant models. Conclusions: The proposed 2018 WCRF/AICR Score is a practical tool for researchers to examine how adherence to the 2018 WCRF/AICR Recommendations relates to cancer risk and mortality in various adult populations.

## 1. Introduction

In 2018, the World Cancer Research Fund (WCRF) and American Institute for Cancer Research (AICR) published Diet, Nutrition, Physical Activity, and Cancer: A Global Perspective, the WCRF/AICR Third Expert Report [1]. Given that an estimated 30–50% of all cancer cases are preventable, this report underpinned the updated ten Cancer Prevention Recommendations that set the benchmark for evidence-based guidance, including eating a healthy diet, maintaining a healthy body weight, and engaging in regular physical activity [1]. An important point made by the WCRF/AICR Continuous Update Project (CUP) Expert Panel, who authored the Recommendations, was that each recommendation was intended to be one part of a comprehensive package of modifiable lifestyle behaviors that, when taken together, promote a healthy pattern of diet and physical activity conducive to the prevention of cancer, other non-communicable diseases, and obesity. Since there is strong evidence that greater body fatness is associated with a number of cancers, the recommendation “Be a healthy weight” appeared first. There are also two targeted recommendations, one for breastfeeding women and one for cancer survivors (Table 1). 

Many studies used the 2007 WCRF/AICR Cancer Prevention Recommendations to derive scores used to investigate associations between adherence and both cancer risk and health outcomes after a cancer diagnosis [2,3,4,5,6,7,8,9,10,11,12,13,14,15,16,17,18,19,20,21]. However, no standard scoring approach was previously developed to define adherence to the WCRF/AICR Recommendations, so each study derived their own version. The previously published scores vary considerably both in the cut-points used and the number of recommendations operationalized. They all used a points-based system; typically, meeting the recommendation scored a full point and, in some cases, partially meeting a recommendation received a half point. Failing to meet the recommendation conferred a zero score for that component. The available scoring approaches included six to nine of the possible 10 recommendations, resulting in scores ranging from zero to nine points. Findings show that greater adherence to the 2007 WCRF/AICR recommendations is associated with reduced risk and incidence of general and site-specific cancers [3,4,5,7,8,9,10,12,14,15,16,20,21], as well as reduced all-cause and cancer-specific mortality [6,11,13,17,18,19,22]. 

Although these studies provide valuable overall support for the benefits of adherence to the 2007 WCRF/AICR Cancer Prevention Recommendations, the ad hoc creation of previous scoring approaches limits the direct comparability of the study results. Given the recent release of the 2018 WCRF/AICR Third Expert Report and updated Cancer Prevention Recommendations, this is an opportune time to address these inconsistencies and provide a framework for greater consistency and comparability of future studies. Thus, this paper provides results of a collaboration between researchers at the U.S. National Cancer Institute (NCI), and members of AICR and WCRF International, in consultation with the WCRF/AICR CUP Expert Panel and additional international researchers, to develop a standard scoring system.

## 2. Materials and Methods 

### 2.1. Score Participants and Approach

Researchers at NCI (Marissa M. Shams-White, Jill Reedy) formed a collaborative team to address the challenge of creating a standardized scoring system that could be operationalized globally. The team included researchers from AICR (Alice Bender, Nigel T. Brockton), WCRF International (Susannah Brown, Panagiota Mitrou), and ISGlobal (Dora Romaguera), the latter of whom was involved in the development of the first reported adherence score based on the 2007 WCRF/AICR Cancer Prevention Recommendations [23]. The NCI-led collaborative team developed the preliminary scoring approach and justifications. We submitted this proposal to the WCRF/AICR CUP Expert Panel for review and feedback on the scoring approach, and received additional guidance from Dr. Teresa Norat, Principal Investigator (PI) of the CUP team at Imperial College London, who was the PI of the project that developed the first 2007 adherence score [15].

### 2.2. Creating the Score

First, we examined the ten 2018 WCRF/AICR Cancer Prevention Recommendations [24] to assess which recommendations and sub-recommendations should be included in the new 2018 WCRF/AICR Score (detailed in Table 1). Given our goal to create a simple, clear scoring system with each recommendation included given equal weighting, we omitted two recommendations. The recommendation to avoid dietary supplement use for cancer prevention and consume nutrients through food alone is largely addressed through the other five dietary recommendations. We also excluded the recommendation specific to cancer survivors to follow the Recommendations after a cancer diagnosis because it would necessarily be derived from a composite measure of the other components of the score and would double-count the impact of adherence. Furthermore, the recommendation on breastfeeding is specific to mothers that are able to breastfeed. Because it is limited to a subgroup of the population and currently linked to a reduced risk of breast cancer in the mother and is protective against weight gain, overweight, and obesity in breastfed children, we retained it as an optional scoring component.

Second, to develop the scoring approach, we conducted a literature review to examine and compare the scoring approaches developed in the previously reported studies examining adherence to the 2007 WCRF/AICR Recommendations [23]. The most commonly considered approaches were binary (1 vs. 0 points for meeting or not meeting each recommendation) and three-level (1, 0.5, and 0 points for meeting, partially meeting, and not meeting the recommendation, respectively) [2,3,4,5,6,7,8,9,10,11,12,13,14,15,16,17,18,19,20,21].

Third, once we selected the recommendations and scoring approach, we considered the factors relevant for defining the cut-points for scoring adherence to each recommendation. The 2018 WCRF/AICR Score utilized all quantitative values that the 2018 WCRF/AICR Expert Report explicitly stated [1]. Where the text of the 2018 Cancer Prevention Recommendations did not provide explicit cut-points, we reviewed guidelines from other leading organizations, such as the World Health Organization (WHO) and the Centers for Disease Control and Prevention (CDC), for guidance. Additionally, we reviewed and considered the precedent available from earlier scores developed to align with the 2007 WCRF/AICR Recommendations [2,3,4,5,6,7,8,9,10,11,12,13,14,15,16,17,18,19,20,21]. Lastly, where limited guidance was available, specifically in relation to the fast-food recommendation, we conducted a literature search in an effort to identify other commonly used scoring systems.

## 3. Results

The proposed 2018 WCRF/AICR Score is summarized in Table 2. The Score is comprised of seven or eight components, depending on the optional inclusion of the breastfeeding component, for a total value ranging from 0 to 7 or 8 points, respectively. We chose a three-level scoring system, striving for a more continuous variable, given that partially meeting a recommendation may confer some benefit (e.g., some physical activity may be more beneficial than no physical activity). Within each recommendation, if more than one sub-recommendation is operationalized, the scoring weight is divided equally between them to retain a total of one point (e.g., two sub-recommendations are worth up to 0.5 points each: 0.5, 0.25, and 0 points for meeting, partially meeting, and not meeting each sub-recommendation, respectively). The details on each Score component are included below.

### 3.1. Healthy Weight Sub-Score

We next sought to operationalize two weight goals to (1) keep weight as low as one can within the healthy range throughout life and (2) avoid weight gain through adulthood (Table 1). For these, we focused on adults and used both body mass index (BMI) and waist circumference (WC). BMI and WC each receive up to half a point. However, if data are only available for either BMI or WC, the sub-score is doubled to retain the 0–1 point total range for the component.

We defined the BMI sub-score cut-points shown in Table 2 based on 2018 WCRF/AICR Recommendations, with further support from WHO and CDC guidance for underweight, normal weight, overweight, and obese. The WC sub-score cut-point for meeting the recommendation is based on 2018 WCRF/AICR Recommendation guidelines, while the cut-points for partially and not meeting the recommendation are based on guidance from the CDC [25] and U.S. National Heart, Lung, and Blood Institute [26]. As indicated in Table 2, to be consistent with the aforementioned guidance, we stratified WC cut-points by sex.

The third goal under this recommendation pertains to body weight in childhood and adolescence (Table 1). It is not included in this sub-score, given the challenges of operationalizing weight gain from childhood through adulthood within the scope of the scoring system proposed.

### 3.2. Physical Activity Sub-Score

The goal operationalized for this recommendation is to be at least moderately physically active and follow or exceed national guidelines. The WHO and U.S. Physical Activity Guidelines (USPAG) advise adults to be active daily and to engage in at least 150 min of moderate-intensity aerobic physical activity or at least 75 min of vigorous-intensity throughout the week [29,30]; thus, we allocated one point to those who complete ≥150 min of moderate-to-vigorous physical activity per week. We based the standard for the 0.5 cut-point on additional data from the USPAG [27], which shows a significantly decreased risk of all-cause mortality even for those who perform 75–<150 min/week of moderate physical activity per week. Moderate-to-vigorous physical activity below 75 min receives 0 points.

We considered a higher level of 300 min or more of moderate-to-vigorous physical activity per week for this recommendation because evidence supports improved health benefits associated with higher volumes and intensity of activity. However, this would penalize those meeting the WHO guidelines and USPAG (i.e., anything ≥150–<300 min/week qualifies as meeting their recommendations).

Although this recommendation also includes a goal to limit sedentary habits (e.g., including screen time), sedentary habits are currently excluded from the proposed scoring system because the 2018 WCRF/AICR Recommendations, USPAG, and current body of literature do not indicate specific minimum values to support the operationalization of this goal. The lack of consensus on cut-points for sedentary habits is supported by evidence from the USPAG that states that the positive association between daily sitting time and all-cause mortality is moderated by volume of physical activity [31].

### 3.3. Wholegrains, Vegetables, Fruit, and Beans Sub-Score

Consistent with previous scores based on the 2007 WCRF/AICR Recommendations, we only operationalized the two goals pertaining to fruit and vegetable and fiber intake (Table 1). For fruit and vegetable and fiber intake, the 2018 WCRF/AICR recommendation to consume at least 400 g and 30 g per day from food sources, respectively, defines the highest scoring standards for each sub-component. We modelled the cut-points for partially and not meeting these recommendations on the consensus of scores based on the 2007 WCRF/AICR Recommendations, due to evidence supporting that meeting at least half of each recommendation is associated with health benefits [2,3,4,5,6,7,8,9,10,11,12,13,14,15,16,17,18,19,20,21].

Although the recommendation also includes two goals that specify wholegrains and legumes, we excluded these given the absence of specific intake goals or standards for both, and their overlap with the fiber goal (i.e., the fiber recommendation is for food sources).

### 3.4. “Fast-foods” Sub-Score

We gave great consideration to the operationalization of the recommendation to limit consumption of “fast foods” and other processed foods high in fat, starches, or sugars, as there is currently no standard way to categorize or define cut-points for it. This recommendation changed significantly from the corresponding 2007 recommendation, which was to limit energy-dense foods by consuming them sparingly, avoiding sugary drinks (now a separate recommendation), and consuming fast-foods sparingly, if at all [23]. Thus, the standards used by researchers who created the 2007-based scoring systems could not be directly applied to the current Score [2,3,4,5,6,7,8,12,13,15,16,17,18]. Through an extensive literature search and consultation with the CUP Expert Panel, we considered various ideas, including examining percent of total kilocalories from saturated fat and/or added sugars. We rejected these due to their focus on two specific nutrients over food sources and the significant contributions from both red and processed meats and sugar-sweetened drinks, respectively. Our solution was to create an adapted version of the NOVA classification system [28] to create an “ultra-processed foods” (UPFs) variable. However, as the NOVA classification system’s definition of UPFs includes foods already accounted for in the 2018 WCRF/AICR Recommendations (i.e., sugar-sweetened drinks and red and processed meats), we created an adapted NOVA classification version to ensure these foods did not receive increased weighting in the scoring approach (i.e., double penalization). Due to the acknowledged challenges of operationalizing this component of the Score, we examined the relationship between previous derivations of the scoring approach and the newly proposed approach. We also examined the percent of total energy per day from the adapted UPF (aUPF) variable in the NIH-AARP Diet and Health (AARP) Study (Appendix A) and the Multi-Case Control Spain (MCC-Spain) Study (data not shown). Among participants in the AARP Study, we found a moderate correlation between percent of energy from aUPF and percent of total energy from saturated fat and added sugars, even after excluding red and processed meat and sugar-sweetened drinks from all variables (data not shown). Similar trends were seen in the MCC-Spain Study (data not shown). A flow chart for this process is depicted in Figure 1.

The cut-points for the aUPF variable were an additional point of debate. Since there is no existing literature regarding the UPF variable to establish standards for the NOVA classification system [32,33,34,35], we decided on subjective cut-points based on tertiles within each specific dataset. This approach accounts for (1) the different UPFs in different countries’ food systems, (2) the variations in focus on UPF food sources in dietary questionnaires (e.g., not unfairly penalizing participants who complete a food frequency questionnaire (FFQ) with more questions on UPFs than another FFQ), (3) the use of a FFQ versus a 24-h recall, and (4) the different approaches researchers may take to quantify UPFs (e.g., percent of total energy versus total grams).

### 3.5. Red and Processed Meat Sub-Score

The goal for those who consume red and/or processed meat is to consume ≤350–500 g red meat/week and “very little, if any” processed meat (Table 1). We defined the latter as <21 g/week (i.e., <3 g/day) to account for the very occasional processed meat consumer and/or recipes that may include minimal processed meat. Thus, we defined fully meeting the recommendation as consuming both <500 g red meat and <21 g processed meat per week; partially meeting the recommendation as consuming either <500 g red meat and 21–<100 g processed meat per week; and not meeting the recommendation as consuming either >500 g red meat or ≥100 g processed meat per week.

We considered using 350 g of red meat per week for the highest standard, but we wanted to account for those still meeting the recommendations from >350–500 g/week.

### 3.6. Sugar-Sweetened Drinks Sub-Score

This recommendation’s goal to avoid sugar-sweetened drinks pertains to any liquids sweetened by adding free sugars and sugars “present in honey, syrups, fruit juices, and fruit juice concentrate.” Therefore, this includes but is not limited to sodas, sports drinks, energy drinks, fruit juices with added sugars, and sweetened water, coffee, and tea beverages [1]. The 2018 WCRF/AICR Score is consistent with most scores based on the 2007 WCRF/AICR Recommendations, whereby we assigned no consumption of sugar-sweetened drinks as meeting this recommendation; ≤250 g/day (1 drink) as partially meeting the recommendation; and >250 g/day as not meeting the recommendation.

### 3.7. Alcohol Consumption Sub-Score

Given the strong evidence that any alcohol consumption increases risk for some cancers [36], the 2018 WCRF/AICR Recommendation, unlike the 2007 version, recommends avoidance of any alcohol [1]. Thus, fully meeting this recommendation requires zero alcohol consumption. However, if alcohol is consumed, the 2018 WCRF/AICR Recommendations support following national guidelines. Thus, in the U.S., males and females who consume no more than 28g of ethanol (2 drinks) and 14 g of ethanol (1 drink) per day, respectfully, partially meet the recommendation; any consumption above this does not meet the recommendation (Table 2).

### 3.8. Breastfeeding Sub-Score

The 2018 WCRF/AICR Recommendation for breastfeeding aligns with the guidance from the WHO, which recommends for infants to be exclusively breastfed for six months, and then for up to two years of age or beyond along with appropriate complementary foods [37]. Thus, cumulative breastfeeding for six or more months meets the recommendation, while any breastfeeding under six months and no breastfeeding partially meets and does not meet the recommendation, respectfully (Table 2).

It is important to note that the inclusion of the breastfeeding component is not mandatory, as it only applies to a specific subpopulation. Its inclusion should be based on the research question and target population and is up to the discretion of the researcher.

## 4. Discussion

Evidence suggests a strong association between engagement in the lifestyle behaviors detailed in the 2018 WCRF/AICR Cancer Prevention Recommendations and cancer risk. We provide a simple, standardized scoring system to quantify adherence to eight of the ten 2018 WCRF/AICR Cancer Prevention Recommendations and to examine how adherence is associated with cancer-related outcomes.

Four of the 2018 WCRF/AICR Score components’ definitions and scoring guidelines partially align with past ad hoc definitions, while four changed significantly (see Appendix A). For bodyweight, many past studies similarly operationalized BMI using underweight, normal, overweight, and obese cut-points [3,6,9,11,13,14,15,16,18,19,20,21], although only one study considered WC [5] and none considered them together. We scored plant foods, fruits, and vegetables similarly to a few past studies [3,5,9,15,16,17], although the fiber cut points changed overall due to the increase in the fiber intake recommendation between 2007 and 2018 from ≥25 g/day to ≥30 g/day. For red and processed meat, many previous studies used similar cut points for red meat [3,5,6,9,12,13,14,15,16,17,19,20,21], but we lowered the cut-off for partially meeting the recommendation for processed meat in the 2018 Score from 21–150 g/week [3,6,9,12,13,14,15,16,17,19,20,21] to 21–<100 g/week. The definition of adherence to the breastfeeding recommendation did not change from the few past studies that included it [3,6,15,16,38]. Conversely, four recommendations significantly changed between 2007 and 2018, thereby leading to deviations from past 2007 scoring approaches. The physical activity recommendation of ≥30 min/day changed to following national guidelines (in the U.S., at least 150 min of moderate physical activity per day); the alcohol recommendation now supports abstention from any alcohol consumption, unlike the 2007 recommendation which allowed moderate consumption. Regarding foods and drinks that promote weight gain, the focus shifted away from a single energy density recommendation to two recommendations that target consumption of “fast foods” and other processed foods and sugar-sweetened drinks separately. The 2007 recommendation on preservation, processing, and preparation was removed from the global recommendations and included as a regional issue relevant in specific parts of the world [24].

As previously mentioned, our decisions on sub-score cut-points were guided by the 2018 Cancer Prevention Recommendations, as well as by guidelines from other leading organizations, past scoring systems, and the existing adherence literature. It is important to further clarify our proposed scoring for the dietary components that are meant to be limited or avoided. The recommendation for alcohol is to “limit alcohol consumption” but this is clarified as “it’s best not to drink alcohol”; therefore, any consumption of alcoholic beverages scores 0.5 or 0 points, depending on servings per day. We decided against a zero limit for processed meat as a requirement for 1 point in the red and processed meat sub-score because processed meats may be included sparingly in some recipes; <3 g/day accounts for these minimal amounts. Given these two approaches, we debated the scoring for sugar-sweetened drinks (i.e., considering if the occasional sugar-sweetened drink should score 1 vs. 0.5 points). Given that the goal is “do not consume sugar-sweetened drinks” (Table 1), abstention receives 1 point. However, we recommend researchers consider how their dietary instrument and nutrient database capture sugar-sweetened drinks when assessing the minimal level of intake permissible for this component.

Although the proposed 2018 WCRF/AICR Score focuses on eight key lifestyle components, it does not address all major risk factors that should be considered when examining cancer-related outcomes. For example, the 2018 WCRF/AICR Cancer Prevention Recommendations include smoking in an overall statement with other cancer risk-related behaviors, such as sun exposure practices, but not as a specific recommendation. Given the major complexities of various smoking variables one needs to consider, depending on the research question and data availability (e.g., smoker versus non-smoker or former smoker, years smoking, number of cigarettes), it is instead strongly encouraged that researchers include smoking and other risk factors in their models when appropriate.

There are a few limitations to the 2018 WCRF/AICR Score. First, our goal was to create a simple overall scoring system, weighting each recommendation equally. It is debatable as to whether weighting should be equal within components (e.g., for calculating the alcohol sub-score in, women, considering if two servings of ethanol/d should be treated the same as 10 servings of ethanol/d) and between components (e.g., giving equal weight to breastfeeding and body weight), and if the weighting should vary based on outcome. Regarding the latter, there is strong evidence linking body fatness to several cancers (hence it appears as the first of the 10 Recommendations), but it is not weighted in this simple Score; there is also a stronger dose-response relationship between alcohol versus sugar-sweetened drink intake and cancer risk that is not currently addressed in the Score. Furthermore, there are five dietary components in the 2018 WCRF/AICR Score, giving greater weight to diet than body weight, physical activity, and breastfeeding. To address these questions, we will examine the implications of reweighting the components in the aforementioned scenarios (e.g., within components, between components, and between lifestyle factors) and to test the predictive validity of the Score in a future methodological paper.

Second, aUPFs are used in the Score to represent “fast-foods” through an adapted NOVA classification system, though UPFs are not explicitly specified in the 2018 WCRF/AICR Expert Report. There is a lack of established standards both in the report’s recommendation and the cancer literature to capture “fast foods” in the diet. We conducted an extensive literature search and consulted the CUP Expert Panel; reviewed the distribution of percent of total energy from UPFs from past studies [32,33,34,35]; and examined the distribution of percent of energy from aUPF within the AARP Study and found it comparable to the percent of total energy from added sugars and saturated fat. Furthermore, though the “fast-food” component is the only one defining cut-points via data-driven tertiles, this may also be viewed as a strength: the cut-points help to address the variation of the aUPF sources of a given population’s food system and the variation in measurements by researchers (e.g., grams versus %kcals), thereby allowing for a more comparable Score across future studies. The goal is to thus provide this aUPF-based framework to researchers, with the intention that future efforts will further inform our understanding of this construct.

Lastly, we did not operationalize all of the sub-recommendations included in the 2018 WCRF/AICR Recommendations in the current Score. Rather, we based these decisions on either limitations in the literature, lack of specificity in the sub-recommendations, or redundancy with another recommendation (which would thus lead to unequal weighting). For example, as previously mentioned, we did not operationalize the physical activity sub-recommendation on sedentary time, given the lack of guidance related to specific cut-points in the 2007 and 2018 WCRF/AICR Expert Reports, the 2018 USPAG, and limited evidence in the literature available to propose cut-points. The plant-based foods component is another example, where we only operationalized two of the four sub-recommendations (Table 1). This limitation is mitigated in the current Score, as the fiber sub-score captures fiber intake from legume and wholegrain sources and we retained refined grains in the aUPF definition. Nevertheless, future efforts in studies that collect additional data may inform how to better capture these constructs (e.g., through cohorts that assess weight patterns from childhood, objective measures of physical activity and sedentary time, and the consideration of different food-based constructs).

There are many strengths to the proposed 2018 WCRF/AICR Score. First, except for “fast foods,” we used evidence-based cut-points for the creation of every component sub-score. Second, we created the scoring system in collaboration with researchers at NCI, AICR, WCRF International, and ISGlobal. It is important to highlight that our scoring approach was also reviewed and evaluated by the WCRF/AICR CUP Expert Panel and areas of ambiguity were examined and discussed. Additionally, the Score is adaptable to various data collection instruments (e.g., different food frequency questionnaires and 24-h recalls can be used to collect the dietary data for this Score). Lastly, the heterogeneity of the 2007-based scoring systems makes it challenging to compare findings across studies and populations. This proposed, standardized approach will allow for future cross-study and population comparisons. We recognize that other researchers may choose to adapt the 2018 WCRF/AICR Score for their particular study (e.g., operationalize the sub-recommendation on sedentary time), but the goal is for the proposed scoring system and its rationale to provide a template for reporting adherence scores and that any modifications will be well-documented.

Our goal is to support the application of the 2018 WCRF/AICR Score to further examine how adherence to the Recommendations is associated with cancer risk and mortality across various populations. Efforts are underway to examine the association between adherence to the Score and cancer risk and mortality in the AARP Study and the MCC-Spain study. We strongly encourage similar efforts in other studies across the world.

## 5. Conclusions

The proposed 2018 WCRF/AICR Score is a practical tool that operationalizes eight of the ten 2018 WCRF/AICR Cancer Prevention Recommendations, whereby a higher score reflects greater adherence to the Recommendations. We encourage researchers to implement this standardized score in their epidemiologic and clinical studies to enhance comparability of findings across populations and countries. By doing so, researchers can promote a combined effort to examine how adherence to the 2018 WCRF/AICR Recommendations relates to cancer risk and mortality in various adult populations.

## Figures and Tables

**Figure 1 nutrients-11-01572-f001:**
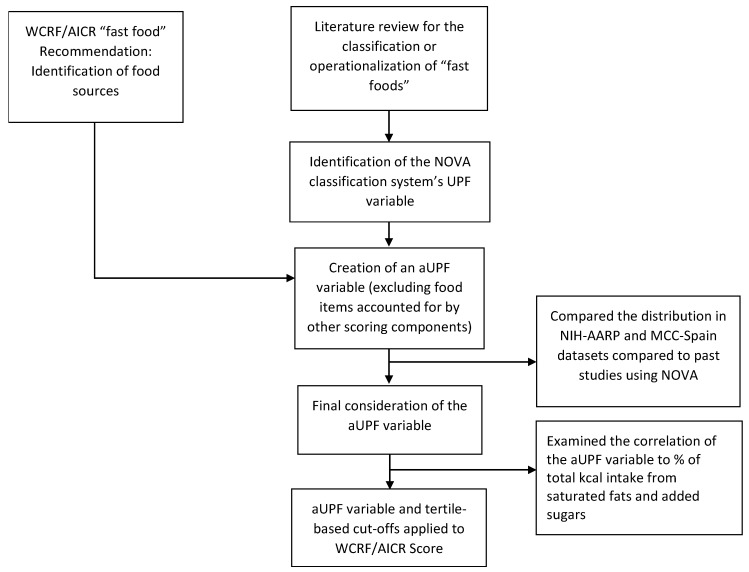
Flow chart: the creation of the adapted ultra-processed foods variable to represent “fast foods”. aUPF, adapted ultra-processed food; AICR, American Institute for Cancer Research; MCC-Spain, Multi-Case Control (MCC-Spain) study; NIH-AARP, National Institutes of Health-AARP (formerly known as the American Association of Retired Persons); UPF, ultra-processed food; WCRF, World Cancer Research Fund.

**Table 1 nutrients-11-01572-t001:** The 2018 WCRF/AICR Recommendations for Cancer Prevention ^1^.

Recommendations	Details	Goals
1. Be a healthy weight ^2^	Keep your weight within the healthy range and avoid weight gain in adult life	Ensure that body weight during childhood and adolescence projects towards the lower end of the healthy adult BMI range Keep your weight as low as you can within the healthy range throughout life ^2^ Avoid weight gain (measured as body weight or waist circumference) throughout adulthood
2. Be physically active ^2^	Be physically active as part of everyday life and life—walk more and sit less	Be at least moderately physically active, and follow or exceed national guidelines ^2^ Limit sedentary habits
3. Eat a diet rich in wholegrains, vegetables, fruit, and beans ^2^	Make wholegrains, vegetables, fruit, and pulses (legumes) such as beans and lentils a major part of your usual diet	Consume a diet that provides at least 30 g/day of fiber from food sources ^2^ Include in more meals foods containing wholegrains, non-starchy vegetables, fruit, and pulses (legumes) such as beans and lentils Eat a diet high in all types of plant foods including at least five portions or servings (at least 400 g or 15 oz in total) of a variety of non-starchy vegetables and fruit every day ^2^ If you eat starchy roots and tubers as staple foods, eat non-starchy vegetables, fruit, and pulses (legumes) regularly too if possible
4. Limit consumption of “fast foods” and other processed foods high in fat, starches or sugars ^2^	Limiting these foods helps control calorie intake and maintain a healthy weight	Limit consumption of processed foods high in fat, starches or sugars—including “fast foods”, many pre-pared dishes, snacks, bakery foods and desserts; and confectionery (candy) ^2^
5. Limit consumption of red and processed meat ^2^	Eat no more than moderate amounts of red meat, such as beef, pork, and lamb. Eat little, if any, processed meat.	If you eat red meat, limit consumption to no more than about three portions per week. Three portions are equivalent to about 350 to 500 g (about 12 to 18 oz) cooked weight of red meat. Consume very little, if any, processed meat. ^2^
6. Limit consumption of sugar-sweetened drinks ^2^	Drink mostly water and unsweetened drinks	Do not consume sugar-sweetened drinks ^2^
7. Limit alcohol consumption ^2^	For cancer prevention, it’s best not to drink alcohol	For cancer prevention, it’s best not to drink alcohol ^2^
8. Do not use supplements for cancer prevention	Aim to meet nutritional needs through diet alone	High-dose dietary supplements are not recommended for cancer prevention—aim to meet nutritional needs through diet alone
9. For mothers: breastfeed your baby, if you can ^2^	Breastfeeding is good for both mother and baby	This Recommendation aligns with the advice of the WHO, which recommends infants are exclusively breastfed for 6 months, and then up to 2 years of age or beyond alongside appropriate complementary foods ^2^
10. After a cancer diagnosis: follow our Recommendations, if you can	Check with your health professional what is right for you	All cancer survivors should receive nutritional care and guidance on physical activity from trained professionals Unless otherwise advised, and if you can, all cancer survivors are advised to follow the Cancer Prevention Recommendations as far as possible after the acute stage of treatment

^1^ AICR, American Institute for Cancer Research; BMI, body mass index; WCRF, World Cancer Research Fund; WHO, World Health Organization. ^2^ This Recommendation or Goal is operationalized in the 2018 WCRF/AICR Score.

**Table 2 nutrients-11-01572-t002:** A Breakdown of the official 2018 WCRF/AICR Score ^1^.

2018 WCRF/AICR Recommendations	Operationalization of Recommendations	Points
1. Be a healthy weight	**BMI (kg/m^2^):** ^2^	
	18.5–24.9	0.5
	25–29.9	0.25
	<18.5 or ≥30	0
	**Waist circumference (cm (in)):** ^2,3^	
	Men: <94 (<37) Women: <80 (<31.5)	0.5
	Men: 94–<102 (37–<40) Women: 80–<88 (31.5–<35)	0.25
	Men: ≥102 (≥40) Women: ≥88 (≥35)	0
2. Be physically active	**Total moderate-vigorous physical activity (min/wk):** ^4^	
	≥150	1
	75–<150	0.5
	<75	0
3. Eat a diet rich in wholegrains, vegetables, fruit and beans	**Fruits and vegetables (g/day):** ^5^	
	≥400	0.5
	200–<400	0.25
	<200	0
	**Total fiber (g/day):** ^5^	
	≥30	0.5
	15–<30	0.25
	<15	0
4. Limit consumption of “fast foods” and other processed foods high in fat, starches or sugars	**Percent of total kcal from ultra-processed foods (aUPFs):** ^6^	
	Tertile 1	1
	Tertile 2	0.5
	Tertile 3	0
5. Limit consumption of red and processed meat	**Total red meat (g/wk) and processed meat (g/wk):**	
	Red meat <500 and processed meat <21	1
	Red meat <500 and processed meat 21–<100	0.5
	Red meat >500 or processed meat ≥100	0
6. Limit consumption of sugar-sweetened drinks	**Total sugar-sweetened drinks (g/day):**	
	0	1
	>0–≤250	0.5
	>250	0
7. Limit alcohol consumption	**Total ethanol (g/day):**	
	0	1
	>0–≤28 (2 drinks) males and ≤14 (1 drink) females	0.5
	>28 (2 drinks) males and >14 (1 drink) females	0
8. (Optional) For mothers: breastfeed your baby, if you can	**Exclusively breastfed over lifetime for a total of:**	
	6+ months	1
	>0–<6 months	0.5
	Never	0
**Total Score Range**		0–7 (or 0–8)

^1^ AICR, American Institute for Cancer Research; aUPFs, adapted ultra-processed foods; BMI, body mass index; kcal, kilocalorie; mo, months; UPF, ultra-processed food; WCRF, World Cancer Research Fund; wk, week. ^2^ Scoring note: When data is available for both BMI and waist circumference, the sum of the two will be used to score. When only one is available, the point values will be doubled to score (i.e., in both scenarios, this subcomponent’s total range will remain 0–1). ^3^ The 1 point cut-point is based on the 2018 WCRF/AICR Recommendation; the 0.5 and 0 pt cut-points are based on Centers for Disease Control and Prevention [25] and National Heart, Lung, and Blood Institute [26] guidelines. ^4^ The 1-pt cut-point is based on the minimum Recommendation. The 0.5 and 0 pt cut-point are based on additional data from the U.S. Physical Activity Guidelines [27]. ^5^ Scoring note: the scoring of this recommendation is consistent with previous approaches used in many past 2007 WCRF/AICR Recommendation-based scores [23], where meeting at least half the Recommendations (i.e., the fruit and vegetable recommendation or the fiber recommendation) earns 0.5 points. ^6^ The UPF variable was created based on the NOVA classification system [28]. Food items already included in other components of the score (i.e., sugar-sweetened drinks, red meats and processed meats) were removed from the original NOVA UPF variable to create the 2018 WCRF/AICR Score adapted UPF (aUPF) variable.

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
