# Peer review of "Operationalizing the 2018 World Cancer Research Fund/American Institute for Cancer Research (WCRF/AICR) Cancer Prevention Recommendations: A Standardized Scoring System"

_nutrients, 2019, doi:10.3390/nu11071572_

Reviewer 1 Report

Dear Authors

In my opinion the theme of the article is very actual and interesting for the readers of the journal.

A collaborative group was formed to develop a standardized scoring system and provide guidance for research applications, following the publication of the 2018 World Cancer Research Fund (WCRF) and American Institute for Cancer Research (AICR) Third Expert Report.

The authors examined the 2018 WCRF/AICR for Cancer Prevention Recommendations, goals, and statements of advice in order to define components of the new score.

The study results show that eight of the ten 2018 WCRF/AICR Recommendations concerning weight, physical activity, diet, and breastfeeding (optional) were selected for inclusion. Each component is worth one point: 1, 0.5, and 0 points for fully, partially, and not meeting each recommendation, respectively (Score: 0 to 7-8 points). Additional guidance stresses the importance of accounting for other risk factors (e.g., smoking) in relevant models.

The authors conclude that the proposed 2018 WCRF/AICR Score is a practical tool for researchers to examine how adherence to the 2018 WCRF/AICR Recommendations relates to cancer risk and mortality in various populations.

The paper is well structured, the title and abstract clearly describe the content of the manuscript, and the language is correct and clear. However, the paper must have a Conclusion section, after Discussion section.

Best regards

Author Response

Point: The paper is well structured, the title and abstract clearly describe the content of the manuscript, and the language is correct and clear. However, the paper must have a Conclusion section, after Discussion section.

Response: Thank you for your review and comment. A Conclusion section is now added to the manuscript (Lines 390-397).

Reviewer 2 Report

This study presented information about the development of a scoring system for those at risk for or who have/had cancer. There are a few recommendations throughout the manuscript to enhance the information:

Abstract:

Background: recommend specifying what the standardized scoring system is for and for the population (adults and children, adults only, etc)

Results: How many total points would be considered a ‘good’ score, 6/7 points or would the total score be more based on the areas within a construct (e.g. diet recommendations may have multiple scores for a total amount in that area)? Please specify

Method:

Aside from the reviewers creating a scoring system, explain how this scoring system was validated and ensured to be reliable. Also, explain how analysis was run on previous data sets to ensure it is an acceptable scoring system.

For those readers unfamiliar with the 2007 scoring system and even this newly developed scoring system, it would be good to explain how this data is collected or expected to be collected, especially for the diet intake. For example, if using the NCI dietary intake and someone indicates that over the past 6 months or 1 year they consumed juice at least once per day, do they receive a score of 0? Also, is the information about whole grains, fruits, and vegetables compiled to identify 1 score? For the weight score, how is it identified that an individual kept a healthy weight throughout their life, if focusing on adults?

Line 126: specify the other commonly used scoring systems for fast food consumption

Results:

Line 129: Recommend rewording that sentence to be “there are 8 total areas/topics with 1 being a conditional area dependent on if the individual is breastfeeding”

Table 2: 4. Limit consumption of fast foods, tertiles 1-3, clarify the kcal percentages that fit within each of these tertiles to indicate if an individual has consumed too much of the processed foods.            

Author Response

Abstract:

Point 1: Background: recommend specifying what the standardized scoring system is for and for the population (adults and children, adults only, etc)

Response 1: This is now specified in the last line of the abstract in Line 40 (in various adult populations).

Point 2: Results: How many total points would be considered a ‘good’ score, 6/7 points or would the total score be more based on the areas within a construct (e.g. diet recommendations may have multiple scores for a total amount in that area)? Please specify

Response 2: Since scoring is currently relative, there isn’t a ‘good’ score we can specify in this way. The newly added Conclusions section now clarifies that a higher 2018 WCRF/AICR Score reflects greater adherence to the 2018 WCRF/AICR Recommendations Conclusion (Lines 392-393).

Method:

Point 3: Aside from the reviewers creating a scoring system, explain how this scoring system was validated and ensured to be reliable.

Response 3: The goal was not to validate the proposed 2018 WCRF/AICR Score, but rather to operationalize and reflect the 2018 WCRF/AICR Recommendations to provide a standardized approach for researchers to allow results to be more comparable across studies and populations. Additional efforts are underway to examine predictive validity and underlying issues around weighting and scoring, as is now fully addressed in Lines 330-343.

Point 4: Also, explain how analysis was run on previous data sets to ensure it is an acceptable scoring system

Response 4: The 2018 WCRF/AICR Score was not run in any data sets, because the scoring system was not created to be based on one data set (which could limit generalizability to other data sets), but rather the guidance in the 2018 WCRF/AICR Third Expert Report. The only component examined in two data sets (one in the US and one in Spain) was ‘fast foods,’ given the limited guidance in the Report, from other leading organizations, and in the literature on how to operationalize it.

Point 5: For those readers unfamiliar with the 2007 scoring system and even this newly developed scoring system, it would be good to explain how this data is collected or expected to be collected, especially for the diet intake. For example, if using the NCI dietary intake and someone indicates that over the past 6 months or 1 year they consumed juice at least once per day, do they receive a score of 0?

Response 5: Thank you for your comment- a sentence was added (Lines 376-377) stating a strength of this Score is that it is adaptable to various data collection instruments. In response to your example, adapting questionnaire response options to address the scoring component is dependent on the instrument used, what the dietary instruments’ response options are and how they are converted, and, where unclear, up to the expertise of the researchers. For example, is the juice on the questionnaire 100% fruit juice or considered a sugar-sweetened juice? If the latter, if the response option is converted by the selected nutritional database to be no more than 8.5 oz per day, then the sub-score would be a 0.5; if the questionnaire response is more than 8.5 oz per day, the sub-score is 0.

Point 6: Also, is the information about whole grains, fruits, and vegetables compiled to identify 1 score?

Response 6: This is correct, similar to past 2007-based scores. Given these goals all fall under the same Recommendation (as shown in Table 1 on Line 107), they are operationalized under one sub-score.

Point 7: For the weight score, how is it identified that an individual kept a healthy weight throughout their life, if focusing on adults?

Response 7: We clarified that the focus is only during the adult lifespan (Line 159).

Point 8: Line 126: specify the other commonly used scoring systems for fast food consumption.

Response 8: The language on Line 126 is now updated to state that “we conducted a literature search in an effort to identify other commonly used scoring systems.” Though we searched for other scoring systems, aside from the NOVA classification system there were unfortunately no other commonly used scoring systems that we could find that operationalized fast food consumption as part of one’s diet.

Results:

Point 9: Line 129: Recommend rewording that sentence to be “there are 8 total areas/topics with 1 being a conditional area dependent on if the individual is breastfeeding”

Response 9: Thank you for your comment. We edited the sentence to remove the parentheses and improve its flow (Lines 129-130). We clarified the text in this way instead of with the recommended rewording, as the breastfeeding component’s inclusion is not only dependent on if the individual is breastfeeding. For example, some cohorts may not have breastfeeding questions in their questionnaires and/or if there are both males and females in the data set, researchers may choose to remove the breastfeeding component to have a single score version for their study.

Point 10: Table 2: 4. Limit consumption of fast foods, tertiles 1-3, clarify the kcal percentages that fit within each of these tertiles to indicate if an individual has consumed too much of the processed foods.

Response 10: We are unable to clarify the kcal percentages that fit into these tertiles, as the nature of tertiles makes the cut points subjective and based off the data set being analyzed.